# The Acoustic Characteristics of Hellenistic Morgantina Theatre in Modern Use

**Giovanni Amadasi** [1], **Antonella Bevilacqua** [2] , **Gino Iannace** [3,*] and **Amelia Trematerra** [3]

1 SCS-ControlSys—Vibro-Acoustic, 35011 Padova, Italy; g.m.amadasi@scs-controlsys.com
2 Department of Industrial Engineering, University of Parma, 43124 Parma, Italy; antonella.bevilacqua@unipr.it
3 Department of Architecture and Industrial Design, University of Campania "Luigi Vanvitelli", 81031 Aversa, Italy; liatrematerra@libero.it
* Correspondence: gino.iannace@unicampania.it

**Abstract:** Thousands of theatres were built during the Hellenistic period in Greece and overseas colonies. The main elements of the Hellenistic theatre are the following: the orchestra, where music and songs were performed to accompany the acting performance, and the *koilon,* where the audience sat. Hellenistic theatres were built without any ceiling, with an open-air configuration. This paper reports the acoustic characteristics of the Greek (Hellenistic) theatre located in Morgantina (Sicily, Italy) based on the technical data gathered in different listening positions selected across the sitting area (*koilon*). The theatre of Morgantina was built in the third century BC and renovated a few decades ago. Nowadays, it is the center of important social and cultural activities. The outcomes of the beamforming technique employed for the survey have been discussed in comparison with traditional acoustic parameters, such as ISO 3382. The scope of this article is to assess the usability of this theatre intended to be used for different types of artistic performances.

**Keywords:** Greek theatres; Sicily; acoustics; beamforming; acoustic measurements; ancient theatres; Morgantina





## 1. Introduction

The first theatres built in ancient Greece were located on the slope of a hill and had a structure with linear steps [1,2]. The Greek theatre is tragedy's place of origin, as testified by the scenes drawn on many vases discovered and preserved in museums. The theatrical performances were accompanied by music, rhythm, and coral dance. Over time, the theatres were transformed into a concentric and geometrical stepped structure to improve the view for all the spectators and to distribute the sound over the audience more uniformly. Nowadays, the reconstruction of these ancient theatres is not very faithful compared to the original shapes because many theatres have been recreated by analogy and comparison with other theatres built in the same period. Due to this methodology, the reconstruction has not always preserved the original geometry and materiality, producing a significant impact on the acoustic response of these places [3]. Vitruvius, in the first century BC, described in the treatise *De Architectura* some general rules to respect the harmony of constructions developed by the Romans; the fundamental principles for achieving correct visual and listening conditions in an ancient theatre are introduced in Book V [4,5] of his manuscript. The rules written by Vitruvius regard the alignment of steps for both acoustic and visual purposes and the installation of vases (*ekeia*) under the marble seats for acoustic correction. According to Vitruvius, the vessels were made of bronze, but in some cities of the *Magna Grecia* in Southern Italy, terracotta vessels have been found [6]. Some of these general principles are considered for modern applications, like the use of resonators for the acoustic correction of rooms.

Before the era of Vitruvius, some research studies hypothesized that the development of the spatial configuration of these performative spaces could have contributed to modeling the field of geometry [7].

Before the curvilinear shape of the Greek theatre, spectators were used to sitting on the natural slope of the hill (*theatron*) characterized by any shape. Originally, the first theatres had steps arranged in a straight line in the shape of a trapezoid facing the position of the actors. To have an increasingly spacious stage environment, improve the vision of the shows, and have the actors in a frontal position with respect to the spectators, according to a direct source–receiver line, they began to build theatres with a circular and elongated plan with several levels of steps [8]. In the Hellenistic period, the seating area built with wooden planks was called *ikria*; the name *koilon* was given to identify the stone blocks placed in a specific geometric arrangement. The *orchestra* was always used to identify the place dedicated to the performance.

In the first Greek theatres, the stage was delineated by the *orchestra,* and the *koilon* had an undefined geometry and sometimes included wooden benches; thereafter, the *koilon* assumed a geometry, as it is possible to admire in some theatres, with the substitution of wooden benches with stone blocks displaced on the slope of a hill. There is also a hypothesis that ancient Greek theatres were also used for political meetings in which all citizens could participate.

The scenic building was composed of a panel behind which the actors could change their masks; this wooden panel was usually painted to imitate the environment in which the theatre was located [7]. In some representations painted on vases, the main Greek theatre performances involved different characters and various musical instruments (e.g., flute, lyre, zither) [9]. In addition, there were actors wearing masks, musicians, singers, and dancers [10]. The *orchestra* was circular, and the *koilon* was acoustically used for housing the choir, singing, and dancing. The orchestra had a circular plan because, originally, the theatrical performances took place on threshing floors where the oxen ground the wheat following a circular path [11]. Figure 1 shows the main elements composing ancient theatres during the Hellenistic period.

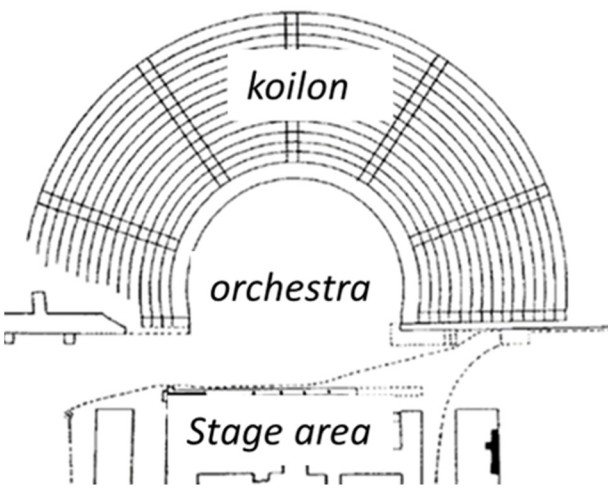

**Figure 1.** Main architectural elements of a Hellenistic theatre.

The adaptation of the other elements surrounding the *orchestra* was gradually developed based on the requirements of the performance.

Starting from a historical and geometric description, this paper deals with the analysis of acoustic characteristics related to a specific Greek (Hellenistic) theatre located in Morgantina (Sicily, Italy). The measured data have been assessed based on the position of the receiver placed in different locations across the sitting area. The theatre of Morgantina was built in the third century BC and renovated only a few decades ago. Nowadays, the theatre is the center of important social and cultural activities.

## 2. Historical Background of Morgantina

In the third century, the city of Morgantina (actually Serra Orlando, Sicily, Italy) reached its greatest splendor under the rule of the tyrant Hieron II. The construction of the theatre was initially attributed to the rich citizen Archela, who consecrated the theatre to Dionysius in the 3rd century BC, as stated on a discovered inscription that occurred in the same place as a pre-existed theatre of smaller size. Some archaeological discoveries in 1956 and 1959 brought to light two pieces of a wall. In 1960, Erik Sjöqvist discovered the extension of the theatre and the remains of a posthumous scenic building [12]. Based on these research studies, the hypothesis that the theatre of Morgantina could be built under the dominion of Hieron II (305–215 BC) prevailed upon the initial discoveries that attributed the construction period to 310 BC.

The theatre of Morgantina was part of a complex connected to the agora, and the *koilon* had a diameter of 57.7 m, which was divided into two sectors: a lower one, consisting of 16 tiers of seats, and an upper one, in clay. The theatre rests on a sloping clearing of the rocky ridge, which was reinforced with infill material (sand and earth). This material was contained by structural walls, the weight of which was supported using buttresses [13]. The entire *koilon* was built with blocks of local limestone, whose geometry was originally trapezoidal with three tiers of steps, later on, transformed into semicircular with additional steps. The orchestra has a diameter of 14.4 m and is paved in a beat. In the following centuries, a long decline for the city of Morgantina began such that the Greek theatre was rediscovered only in 1950. The sitting area for the audience consists of 16 steps (height 0.34 m and width 0.70 m). The sitting area is divided into six wedged sectors using seven radial stairs. Figure 2 shows the view of the theatre in its current state after the latest restoration works in 2005 [14,15].

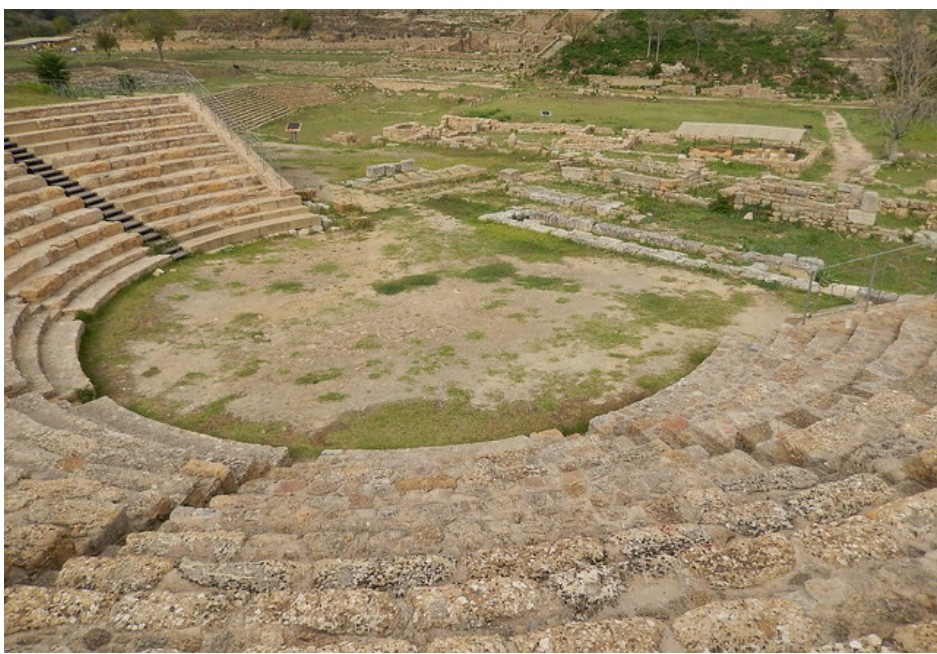

(**a**) orchestra

**Figure 2.** *Cont.*

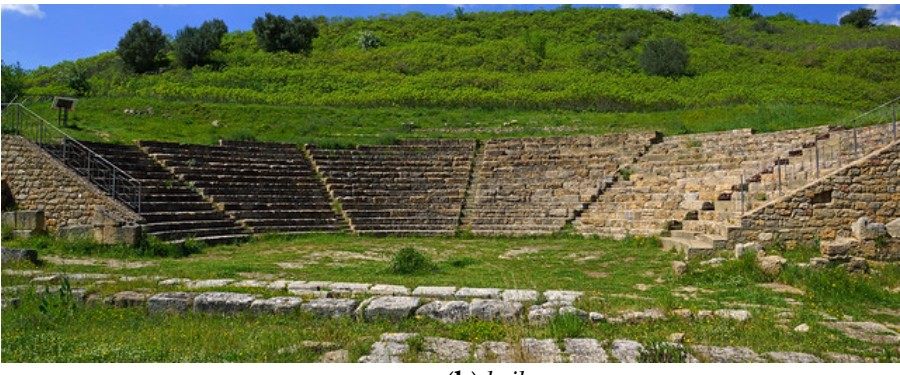

(**b**) *koilon*

**Figure 2.** Views of the archaeological site of Morgantina: (**a**) orchestra, (**b**) *koilon*.

### 3. Acoustic Measurements

A campaign of acoustic measurements has been carried out in the Greek theatre of Morgantina in order to acquire information regarding the acoustic responses of the theatre and analyze the main acoustic parameters in accordance with the standard requirements stated in ISO 3382 [16].

The equipment used during the measurements is listed as follows:
Equalized omnidirectional sound source (B&K Omni Power 4292-L).

- 2-channel microphone (01 dB-Symphonie).
- Dummy head.
- 31-channel spherical array microphone.
- Prismatic array camera (No. 12 lenses).

The sound source was placed above the orchestra area, where a posthumous scenic building could be built, at the height of 1.6 m from the ground, while the receiver was moved across the audience area, in a few selected positions, at 0.8 m from the relative steps height, corresponding approximately to the height of the human ears in a seated position [17]. The sound source was fed using the maximum length sequence (MLS) characterized by 65,535 samples, with a number of slots in the shift register equal to 16 and generated by the 01 dB Symphonie system of 5 s duration. The acoustic parameters taken into consideration are the Early Decay Time (EDT), the reverberation time ($T_{30}$), the music clarity ($C_{80}$), the definition ($D_{50}$), and the Speech Transmission Index (STI) for speech intelligibility [18–21].

The acoustic measurements were taken in unoccupied conditions to keep the background noise at the minimum levels and increase the signal-to-noise ratio (S/N) [22]. The theatre nowadays is located inside the archaeological park, away from road traffic lines and residential properties, with ideal conditions of no pollution from environmental noise. The position of the equipment during the survey is shown in Figure 3.

A brief description of the main acoustic parameters is given as follows [23–29]:

- The early decay time (EDT) is related to the early reflections occurring in the first 10 dB decay after the impulse. The integrated Schroder curve suggests that the EDT can be calculated from the reverberation decay curve. The optimal values for the enclosed theatres are between 1.8 s and 2.6 s.
- Reverberation time refers to the time it takes for sound energy to decay by one millionth compared to the initial level of a stationary sound source. The decay range considered in this research study is between −5 dB and −35 dB ($T_{30}$). The reverberation time is highly dependent on absorption coefficients of the room boundaries other than on the volume of the considered space.
- The clarity index ($C_{80}$) is based on the ratio between the sound reflections arriving within 80 ms (for music) and the sound energy arriving in the following instants of the

soundwave. The optimal value of the clarity index is 0 dB, meaning a perfect balance of early and late reflections, but a tolerance of $\pm 2$ dB is given.

- Definition ($D_{50}$) is another acoustic parameter used for describing the characteristics of a room. The speech definition is targeted between 50% and 100%, while the best range for music is defined between 0% and 50%.
- The speech transmission index (STI) is a parameter for the objective definition of speech intelligibility in a room. The STI parameter ranges from 0 to 1 (0 = bad understanding; 1 = excellent understanding). When the STI is close or at 1, the environment is very suitable for speech understanding. STI aims to objectively quantify the comprehensibility of speech in a specific position in a room, and it is classified in the following ranges: bad < 0.32; poor 0.32–0.45; fair 0.45–0.60; good 0.60–0.75; excellent > 0.75.
- The interaural cross-correlation (IACC) is a binaural acoustic parameter indicating spatial involvement in a certain environment. High values of IACC correspond to a monoaural response, lacking the stereophony and spatial information, while low values of IACC mean a good audio immersion inside the space.

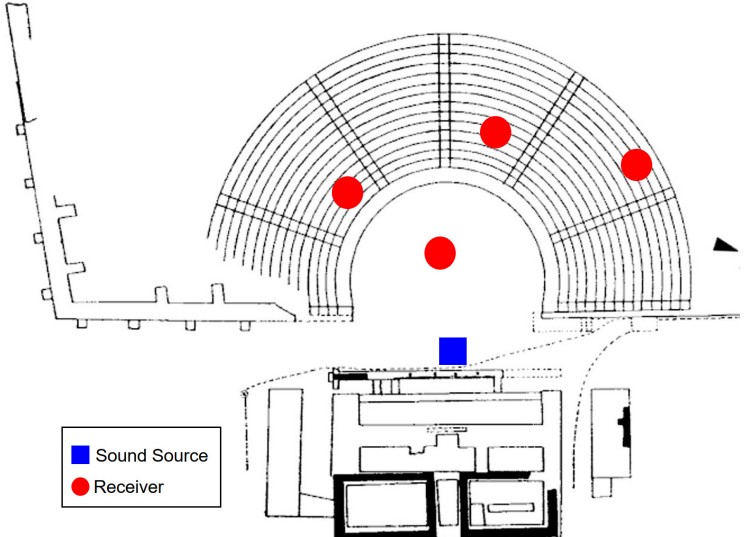

**Figure 3.** Schematic positions of the equipment during the acoustic measurements.

Table 1 summarizes the optimal values of the main acoustic parameters with respect to the room volume and function of the space [30–32].

**Table 1.** Optimal ranges related to the main acoustic parameter for the different listening conditions.

| Parameters | EDT, s | $T_{30}$, s | $C_{80}$, dB | $D_{50}$ |
|---|---|---|---|---|
| Value for listening to the music | $1.8 < EDT < 2.6$ | $1.6 < T_{30} < 2.2$ | $-2 < C_{80} < 2$ | <0.5 |
| Value for speech comprehension | 1.0 | $0.8 < T_{30} < 1.2$ | >2 | >0.5 |

## 4. Results of Monoaural Acoustic Parameters

The analysis of the data in the post-processing phase has been assessed in the bandwidth comprised between 125 Hz and 4 kHz, as required by the standard ISO 3382 [16]. The results shall be intended to be averaged for all the sources and receivers' positions assumed during the acoustic measurements, as shown in Figure 4.

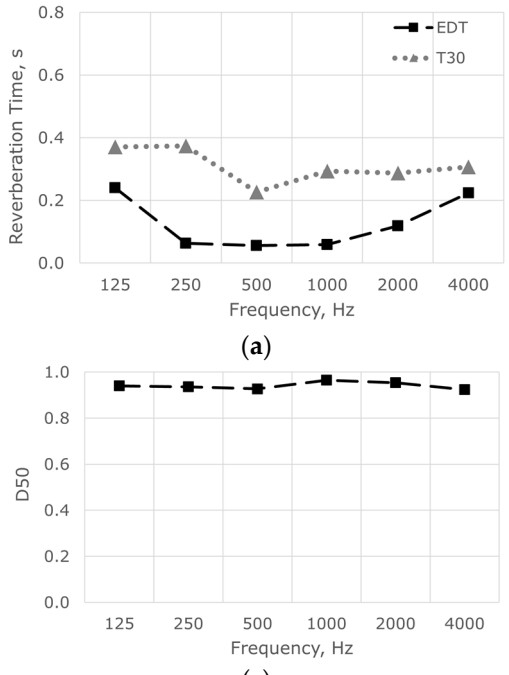

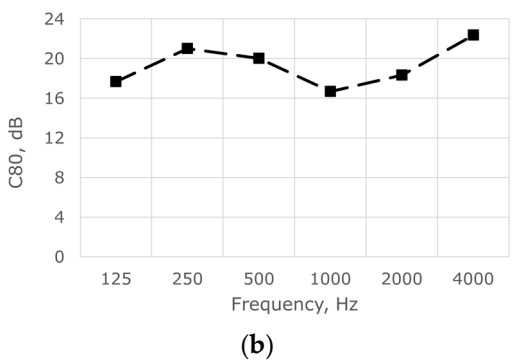

**Figure 4.** Measured results of early decay time and reverberation time (**a**), music clarity (**b**), and definition (**c**).

Figure 4a shows that the EDT values in Morgantina are around 0.1 s at mid frequencies, with an upward trend at 125 and towards 4 kHz. The values related to $T_{30}$ have been found to be equal to 0.3 s if averaged across the spectrum. This means that the theatre of Morgantina, as an open-air theatre, is an acoustically dry environment, which is not suitable for any musical performance.

It can be noticed that the measurements have been carried out without any audience and, therefore, with reflective steps. With the audience, the reverberation time would decrease further, and the effects of sound diffusion from the seating blocks would be reduced. However, these values are comparable with the results obtained from other acoustic measurements carried out in similar open-air theatres. Unfortunately, ancient theatres that are used for modern purposes are characterized by weak reverberation due to the absence of reflecting vertical surfaces [33–35].

A vertical surface behind the orchestra should be placed to improve the reverberation, as this was the function of the scenic building developed later by the Romans in their theatres. In the absence of any vertical surface behind the performing arts space, the sound is not addressed toward the audience and dispersed in other directions.

In terms of clarity, the results of $C_{80}$, as shown in Figure 4b, are around 18 dB, which is far above the upper limit threshold set for good music clarity (i.e., +2 dB), meaning that the music is perceived too clear.

The definition ($D_{50}$), as shown in Figure 4c, is around 0.95 (95%), meaning that the acoustic response is suitable for speech only and not for musical events.

The STI average value has been found to be equal to 0.7, falling into the "good" category. In order to assess the acoustic characteristics of Morgantina theatre, a comparison was conducted with the acoustic parameters measured inside the theatres of Benevento, Pompeii, Posillipo, Taormina, Segesta, Tyndaris, and Siracusa. The average values of the main acoustic parameters, related to the mid-frequency bands of 500 Hz and 1 kHz, are reported in Table 2 as the values measured in unoccupied conditions.

**Table 2.** Measured acoustic parameters averaged at mid-frequency bands of 500 Hz and 1 kHz, related to some ancient theatres.

| Theatres | $T_{30}$ (s) | $C_{80}$ (dB) | $D_{50}$ | *Cavea/Koilon* Diameter (m) |
|---|---|---|---|---|
| Morgantina | 0.3 | 18 | 0.95 | 58 |
| Benevento | 0.9 | 8.0 | 0.78 | 93 |
| Cassino | 0.6 | 19.0 | 0.91 | 53 |
| Pompeii | 0.9 | 6.0 | 0.70 | 60 |
| Posillipo | 1.1 | 3.0 | 0.70 | 47 |
| Taormina | 1.9 | 1.2 | 0.53 | 110 |
| Segesta | 0.5 | 16.0 | 0.90 | 63 |
| Tyndaris | 0.55 | 18 | 0.95 | 76 |
| Siracusa | 1.2 | 13.0 | 0.90 | 140 |

Table 2 shows the measured results of acoustic parameters carried out inside some open-air theatres, which are continuously used for live venues during the summer season. Some theaters included in this comparison have Greek origins, while others were built or modified during the Roman Empire. During the Middle Ages, all these theaters were abandoned due to the barbaric invasions and were transformed or surmounted by residential properties. Only in the last few decades, these theatres have been rediscovered. The reconstruction of some parts of the damaged theatres is carried out using cement instead of stone or tuff blocks, which explains by itself the results of the acoustic characteristics as shown in Table 2. In comparison with others, the theatre in Morgantina has the shortest reverberation time.

## 5. Results of Binaural Acoustic Parameters and Beamforming Mapping

The analysis of the IACC has been conducted with a dummy head and the spherical array beamforming microphone. For all the measuring positions, the IACC value was about 100 ms or lower, meaning that the architectural reflections do not alter the sound perception. Based on the availability of the equipment employed, it is possible to compare the response gathered with the dummy head and the acoustic maps obtained by processing the response recorded with the spherical array microphone. The audio response has been overlapped with the video recorded by the cameras so that it is possible to visualize the reflections resulting from the interaction between sound waves and architecture. With this technique, it is possible to visualize the direct sound and the other rays delaying in time scale hitting the probe of the microphone.

By correlating the sound propagation of sound in the air, assumed to be 344 m/s without any environmental effect of wind and temperature variance, it is possible to trace the architectural reflections inside the theatre with the use of a beamforming sensor characterized by a spherical baffled (close to a sphere), as shown on Figure 5.

For this type of measurement, the position of the spherical array microphone has been taken onto the third step of the *koilon*, starting from the orchestra level, along the central stairwell that divides the theatre into two symmetrical halves, as shown in Figure 7.

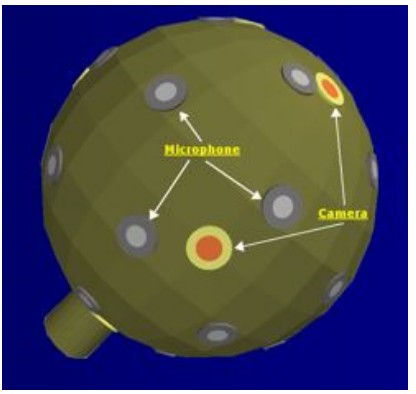
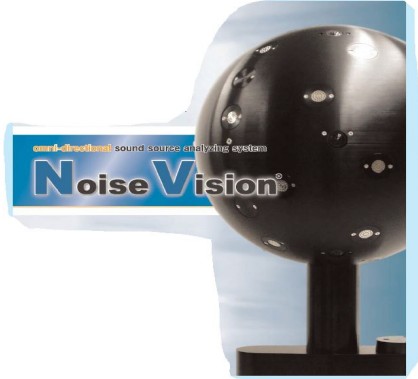

**Figure 5.** Spherical baffled beamforming sensor with 31 transducers and 12 cameras.

For practical reasons, the image of the theatre has been divided into ten areas, as indicated in Figure 6.

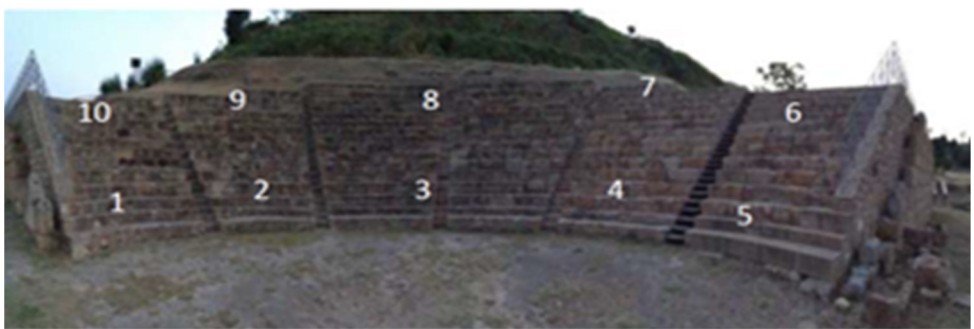

**Figure 6.** Schematic subdivision of the beamforming analysis to identify the reflection areas.

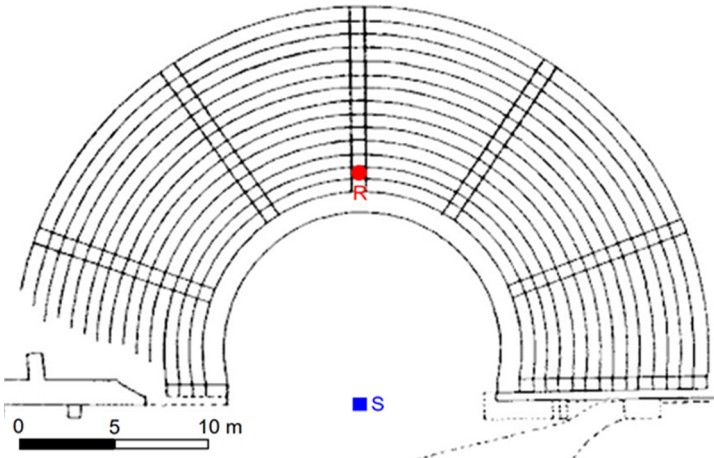

**Figure 7.** Position of the beamforming spherical sensor (R) and the sound source (S) during the beamforming investigation regarding the strength of the sound energy reflections.

The analysis was carried out for the octave bands comprised between 250 Hz and 4 kHz. For 250 Hz, the cross-correlation function is indicated in Figure 8, where it is possible to detect some peaks, as indicated in the lower graph. The visible peaks on the graphs report the time delay between the sound source acoustic waves (direct wave) at the beamforming sensor and the time of arrival of several reflections from the surrounding surfaces. Each peak is a reflection valued in milliseconds and corresponds to the distance of a reflecting surface from the sensor; assuming that the speed of sound is 344 m/s, the distances of 2 m, 7.2 m, and 17.4 m can be calculated to correlate the beamforming results.

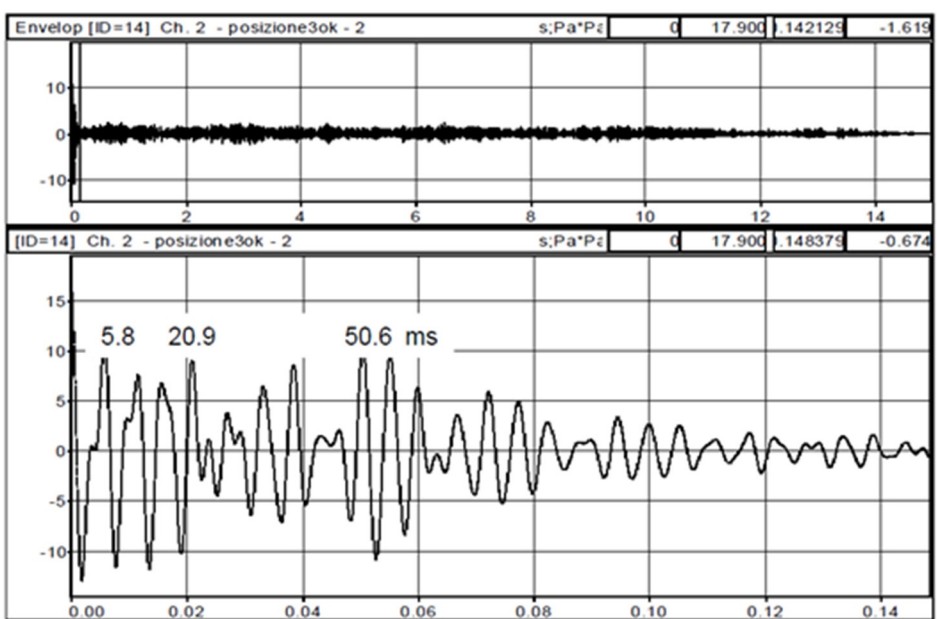

**Figure 8.** Cross-correlation of the sound signal at 250 Hz, received by the beamforming sensor.

The following images, as indicated in Figure 9, refer to 8 of the 12 cameras installed during the survey. It is possible to notice that camera No. 4, 1, 2, and 3 lie on the horizontal measuring plane and cuts the spherical transducer into two parts. The other four cameras (i.e., No. 12, 9, 10, and 11) are pointed downwards, as well as the four left cameras pointing to the sky have not been included since no sound energy has been detected hitting the ground.

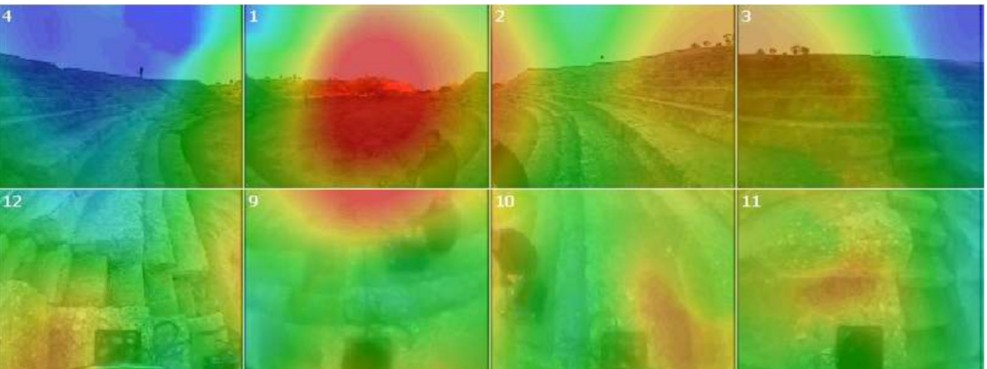

**Figure 9.** Sound energy color map at 250 Hz.

The chromatic scale cartography for the 250 Hz band reveals that the sound energy has been detected in camera No. 1, which is frontally facing the sound source in the proscenium, while a reflection from the right side is visible in camera No. 3, which is directed towards the steps. By comparing these acoustic maps with the cross-correlation curve shown in Figure 9, the reflection can be traced back to the peak at 50 ms, where the path is given by the distance of 16 m (8 m × 2 m), while in-camera No. 10 there is a less marked reflection that can be correlated to the peak placed at 20 ms. However, it can also be noticed that the peaks are distant from each other at a fairly constant time, corresponding to the distance between the rings of the steps.

By zooming on camera No. 3, the area behind the listener is shown in Figure 10. The sound energy map related to this view allows the observation of the reflections' behavior once the soundwave hits the steps of the *koilon* before being recorded by the transducers facing the steps. It can be understood why the feedback on the curve of the cross-correlation is the sequence of peaks with nearly constant time intervals.

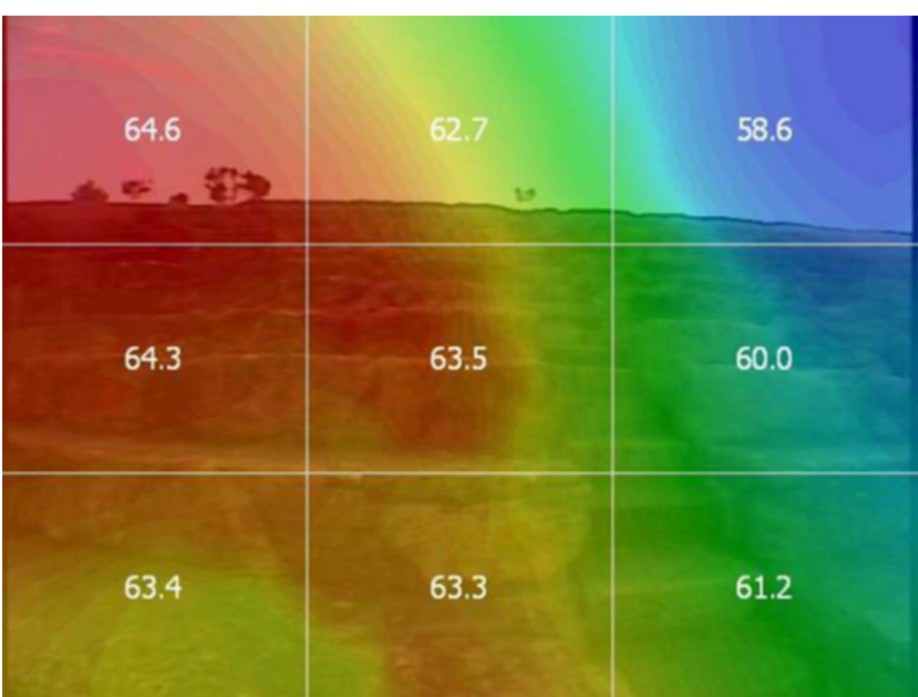

**Figure 10.** Sound energy reflections were detected towards the *koilon* at 250 Hz. The white numbers indicate the sound pressure level (dB) in each segmented area.

## 6. Discussions

This paper discusses the acoustic responses, measured with different equipment and methodologies, inside the ancient theatre of Morgantina. The measured results indicate that, in the actual conditions, the theatre is not suitable for musical performance, given the drier acoustic environment resulting from the lack of vertical surfaces (e.g., scenic building, modern acoustic shell) that can address the sound energy towards the sitting area instead of being dispersed in other directions. However, these results indicate that the open-air theatres, in their current state, are mainly suitable for prose instead of music since the direct soundwave is dominant in the absence of any useful reflection other than ground [36–40]. If Greek or Roman theatres were used for music, they needed some reflecting panels (screens) behind the orchestra to improve the acoustic quality. Another alternative to improve the acoustic response of these ancient theatres is the installation of an amplified audio system. Another possible solution to improve the acoustics of the Theatre of Morgantina would be the installation of temporary structures of high-density materials, like PVC sheets, characterized by reflecting features so that the sound reflections can be addressed to the audience.

A second methodology employed to detect the reflections inside the theatre is beamforming, which is possible to apply with the spherical array microphone and the cameras placed on the faces of a dodecahedron. The results obtained using the acoustic maps have been compared to the IACC values, where it is possible to correlate the single peaks of the cross-correlation response curve with the colored images related to each reflection, also confirmed using the estimation of the traveling distance of soundwaves. On top of the above results confirming the classical characteristics of such theatres, an interesting verification was also carried out about the relation of the IACC index with physical reflections detected and described using a spherical beamforming baffle.

Future research studies will be focusing on the comparison of these results with the acoustic maps realized by a spherical array microphone provided with 64 channels, in combination with a 360° camera, which can render a panoramic photo of the environment without any interruption or gaps between views. The theatre of Morganitina should be modernized if it would like to be used for modern functions in order to improve the

experience of the spectators other than to support the effort of the actors/musicians during the live performance.

## 7. Conclusions

This paper reports the acoustic characteristics of the Hellenistic theatre of Morgantina in actual state. The theatre had its maximum splendor during the 4th century BC. Historical vicissitudes of war and looting led to the abandonment of the site, which was discovered only in the last century. The theatre was rebuilt a few decades ago when 16 steps leaning against the hill were rebuilt without a scenic building.

The theatre was rebuilt with stone blocks available locally, and nowadays, it is located within an archaeological park away from traffic lines and residential properties. The acoustic measurements with various techniques have yielded information that the theatre in the current configuration is not suitable for musical performances, and therefore, it is necessary to install an acoustically reflecting shell on the scene or an amplified audio system. Despite the weak acoustics, the public appreciates attending the shows inside ancient theatres as remarkably historical places handed down to posterity.

**Author Contributions:** Conceptualization, G.A., A.B., G.I. and A.T.; methodology, G.A.; software, G.A.; validation, G.A., A.B. and G.I.; formal analysis, G.A., A.B., G.I. and A.T.; investigation, G.A., A.B., G.I. and A.T.; resources, G.A.; data curation, G.A., A.B., G.I. and A.T.; writing—original draft preparation, G.A., A.B., G.I. and A.T.; writing—review and editing, G.A., A.B., G.I. and A.T.; visualization, G.A., A.B., G.I. and A.T.; supervision, G.A.; project administration, G.A., A.B., G.I. and A.T.; funding acquisition, G.A. and A.B. All authors have read and agreed to the published version of the manuscript.

**Funding:** Funding has been provided by the European Union's Joint Programming Initiative on Cultural Heritage project PHE (The Past Has Ears, phe.pasthasears.eu).

**Data Availability Statement:** Not applicable.

**Conflicts of Interest:** The authors declare no conflict of interest.

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
