# Peer review of "The Acoustic Characteristics of Hellenistic Morgantina Theatre in Modern Use"

_acoustics, doi:10.3390/acoustics5030050_

Round 1
Reviewer 1 Report
LINES 46 to 59 - No mention is made of a stage (skene) as part of a Greek theatre. As this is important for the article, some explanation should be given. It seems that in Greek theatres, as is the case of Roman theatres, there was a back wall at the back of the stage and this would allow the reflection of sound. In his article on “Playing places: the temporary and the permanent” (book: The Cambridge Companion to Greek and Roman Theatre, 2007), pages 204-205, Richard Beacham says: "In the course of the [fifth] century [BC] the skênê, previously of wood and canvas, may have been enlarged, made more elaborate in decor and provided with a serviceable roof for the appearance of gods." This means that Greek theatres had a back stage decoration and also possibly a roof. This should be explained because it may influence the type of methodology that should be used
To insist on this idea. In “The Greek Theatre and Festivals” (2007) by Peter Wilson he explains about backstage rooms in the theatre of Aphrodisias (page 60) and in page 92 and following we are told about stage buildings
LINE 61 - "Figure 1. Main architectural elements of the Hellenistic theatre". Figure 3 seems to indicate that there are remains of a skene (stage), but this is not reflected in figure 1 as, instead of a stage we find the word “landscape”. If the remains are not of the stage, then the author should explain what the remains were. None of the authors are archaeologists and, being this an interdisciplinary paper, it seems that one of the authors should be an archaeologists. There has been extensive excavation work for about a century in the site
LINE 77 - Figure 2. View of the archaeological site of Morgantina (Serra Orlando, Sicily, Italy). - Is this the authors’ photograph? The source if published, and if possible the photographer, should be acknowledged, even if the photograph has been made by the authors.
LINE 80 - Figure 3. Plan of Morgantina theatre in actual state - The source of this figure should be properly referred. This comment applies to the other figures too, but in this case it is unlikely that the authors, who are not either archaeologists or architects, have been able to draw it. The source should be acknowledged.
LINES 263-265 - “lack of vertical surfaces (e.g. scenic building, modern acoustic shell) that can address the sound energy towards the sitting area instead of being disperse in other directions …” - Given that the authors have not properly discussed whether the archaeologists have identified a stage or not, the statement that there was a lack of a vertical surface is unverifiable and the conclusions are invalid. Thus, it seems imperative that the authors explain why they think that there was no a wall at the back of the stage. Only if they assume that there was none (which is doubtful, I think), their results will be valid.
LINES 51-52 - English may need some editing: “During the 4th century BC, the placement of the seats dedicated to aristocrats and important people of the society (proedria) close to the orchestra in order to …” There are other places in the article where I felt that the English needed some (minor) editing
Author Response
LINES 46 to 59 - No mention is made of a stage (skene) as part of a Greek theatre. As this is important for the article, some explanation should be given. It seems that in Greek theatres, as is the case of Roman theatres, there was a back wall at the back of the stage and this would allow the reflection of sound. In his article on “Playing places: the temporary and the permanent” (book: The Cambridge Companion to Greek and Roman Theatre, 2007), pages 204-205, Richard Beacham says: "In the course of the [fifth] century [BC] the skênê, previously of wood and canvas, may have been enlarged, made more elaborate in decor and provided with a serviceable roof for the appearance of gods." This means that Greek theatres had a back stage decoration and also possibly a roof. This should be explained because it may influence the type of methodology that should be used
correct
In most cases a Greek Theatre has some skene, often coincident with the Agorà buildings, however we do not have clear and specific information about the skênê in Morgantina Greek theatre. The construction of the theatre was initially attributed to the rich citizen Archela who consecrated the Theatre to Dionysius in the III century. B.C., as stated on an inscription [1], were a smaller theatre already existed.
After some finds in 1956 and 1959, consisting of two pieces of wall, in 1960 the more realistic extension of the theatre and the remains of a scenic building were found, thanks to the work of Erik Sjöqvist of Princeton University, which void the initial hypothesis (310 BC) and it could hypothesize that it was instead built by Hieron II (305 – 215 BC) the Greek tyrant of Syracuse.
The theatre was part of a complex connected to the agora and the cavea has a diameter of 57.70 m , built in limestone, divided into two sectors: a lower one, made up of sixteen tiers of seats, and an upper one, in clay. It rests on a slightly sloping clearing of the rocky ridge, which was reinforced with fill material (sand and earth). This material was contained by the thick walls, the weight of which was supported by buttresses.
LINE 61 - "Figure 1. Main architectural elements of the Hellenistic theatre".
Figure 3 seems to indicate that there are remains of a skene (stage), but this is not reflected in figure 1 as, instead of a stage we find the word “landscape”. If the remains are not of the stage, then the author should explain what the remains were. None of the authors are archaeologists and, being this an interdisciplinary paper, it seems that one of the authors should be an archaeologists. There has been extensive excavation work for about a century in the site
correct
LINE 77 - Figure 2. View of the archaeological site of Morgantina (Serra Orlando, Sicily, Italy). - Is this the authors’ photograph? The source if published, and if possible the photographer, should be acknowledged, even if the photograph has been made by the authors.
correct
LINE 80 - Figure 3. Plan of Morgantina theatre in actual state - The source of this figure should be properly referred. This comment applies to the other figures too, but in this case it is unlikely that the authors, who are not either archaeologists or architects, have been able to drawit. The source should be acknowledged.
- Sear, F. Roman Theatres: An Architectural Study; OUP Oxford: Oxford, UK, 2006
- Sjöqvist, E., Sicily and the Greeks: studies in the interrelationship between the indigenous populations and the Greek colonists, Ann Arbor (Mich.), University of Michigan Press, 1973.
- Stillwell, R. Excavationsat Morgantina (Serra Orlando) 1966: Preliminary Report IX, American Journal of Archaeology, 1967, 71, (3.), doi: https://doi.org/10.2307/501558
LINES 263-265 - “lack of vertical surfaces (e.g. scenic building, modern acoustic shell) that can address the sound energy towards the sitting area instead of being disperse in other directions …” - Given that the authors have not properly discussed whether the archaeologists have identified a stage or not, the statement that there was a lack of a vertical surface is unverifiable and the conclusions are invalid. Thus, it seems imperative that the authors explain why they think that there was no a wall at the back of the stage. Only if they assume that there was none (which is doubtful, I think), their results will be valid.
The measurement are taken in our days not in the past and the goal was to understand the usability for which kind of performance, of Morgantina Greek Thatre as it is today.
Comments on the Quality of English Language
LINES 51-52 - English may need some editing: “During the 4th century BC, the placement of the seats dedicated to aristocrats and important people of the society (proedria) close to the orchestra in order to …” There are other places in the article where I felt that the English needed some (minor) editing
erase
Reviewer 2 Report
The manuscript investigates the acoustic characteristics of Hellenisitc Morgantina theatre. The topic is of interest. However, the main conception seems to be improved.
1, Does the main article focus on a ruin of the historic theatre, or a rebuilt one?
2, In my opinion, although technology has developed, the craftsmans of ancient times were smart. It will be valuable to simulate the acoustic characteristics of the historical restorations (possible ones), which would be very helpful to the archaeologist and historians to gain the understanding of historic architectures or even to recovering the real original ones.
Author Response
The manuscript investigates the acoustic characteristics of Hellenisitc Morgantina theatre. The topic is of interest. However, the main conception seems to be improved.
1, Does the main article focus on a ruin of the historic theatre, or a rebuilt one?
IT focus on the usability of the Theatre as it is today
2, In my opinion, although technology has developed, the craftsmans of ancient times were smart. It will be valuable to simulate the acoustic characteristics of the historical restorations (possible ones), which would be very helpful to the archaeologist and historians to gain the understanding of historic architectures or even to recovering the real original ones.
An acoustic simulation of the theater in Origone will be the subject of a forthcoming paper.
At the moment we don't know exactly the theater as it was in origin. We can only make assumptions
Reviewer 3 Report
The paper centers around the meticulous documentation of the procedure employed to collect technical data regarding the acoustic characteristics of a prototypical Greek (Hellenistic) theater situated in Morgantina, Sicily, Italy. The evaluation encompassed diverse listening positions within the seating area (koilon). A comprehensive campaign of acoustic measurements was meticulously conducted within the Greek theater of Morgantina. The primary objective was to amass data concerning the theater's acoustic response and subsequently analyze its core acoustic parameters. The work is very complicated and comprehensive. Here are some comments for the authors to clarify their claims and may further improve this manuscript.
1. In the introduction section, it is highly advisable to provide a clear rationale for undertaking this study, outlining the specific motivations and objectives.
2. I recommend giving further attention to the improvement of Figure 10. Additionally, Figures 11 and 12 may appear challenging to comprehend, especially with the added background complexity.
3. Figures 10–12's titles should also be rewritten so that readers may more easily understand what you're trying to express.
I suppose that the paper can be published after minor revision.
Author Response
In the introduction section, it is highly advisable to provide a clear rationale for undertaking this study, outlining the specific motivations and objectives.
The objective was to evaluate and classified the usability of the Theatre as it is for different performances
- I recommend giving further attention to the improvement of Figure 10. Additionally, Figures 11 and 12 may appear challenging to comprehend, especially with the added background complexity.
The visible peaks on the graphs report the time delay between the sound source acoustic waves (direct wave) time of detection at beamnforming sensor and the time of arrival of several reflection from the surrounding surfaces. Each peaks is a reflection values in milliseconds and correspond to a distances of a reflecting surface from the sensor; assuming speed of sound of 344 m/a we can calculate distances as 2.0 m, 7.2 m and 17.4 m, which are necessary to correlate the beamforming results.
- Figures 10–12's titles should also be rewritten so that readers may more easily understand what you're trying to express.
done
Round 2
Reviewer 1 Report
The paper fulfils the criteria for publication. Yet, it would be advisable that either the Greek terms or the Latin terms for theatres were used. Cavea is a Latin term, for example
The article may need some minor English editing
Author Response
revised paper as request